# The Effect of Nozzle Temperature on the Low-Temperature Printing Performance of Low-Viscosity Food Ink

**DOI:** 10.3390/foods12142666

**Published:** 2023-07-11

**Authors:** Qiang Tong, Yuxiang Meng, Yao Tong, Dequan Wang, Xiuping Dong

**Affiliations:** 1College of Mechanical Engineering and Automation, Dalian Polytechnic University, Dalian 116034, China; tongqiang.work@outlook.com (Q.T.); myx.work@outlook.com (Y.M.); tong0555@outlook.com (Y.T.); gtr446962334@gmail.com (D.W.); 2School of Food Science and Technology, Dalian Polytechnic University, Dalian 116034, China

**Keywords:** low-temperature forming, low-viscosity ink, printing temperature, formability

## Abstract

Low-temperature food printing technology is used in many fields, such as personalized nutrition, cooking art, food design and medical nutrition. By precisely controlling the deposition temperature of the ink, a food with a finer and more controllable structure can be produced. This paper investigates the influence of nozzle temperature on printing performance via a numerical simulation and experimental research. The results indicate that the ink gradually changed from a granular state to a fLow-characteristic deposition structure when the nozzle temperature increased from 19 °C to 27 °C. When the nozzle temperature exceeded 21 °C, the ink demonstrated excellent extrusion behavior and tended to flow. The widths of the rectangular frame deposition showed no obvious changes and were 4.07 mm, 4.05 mm and 4.20 mm, respectively. The extrusion behavior of the ink showed a structural mutation in the temperature range of 19–21 °C. Its line width changed from 3.15 mm to 3.73 mm, and its deposition structure changed from a grainy shape to a normal shape. Under the influence of different environmental control capabilities, bulk structure deposition demonstrates an ideal printing performance at 21, 23 and 25 °C, and the latter temperature is more suitable in the case of large external interference. The ink flowed violently when the nozzle temperature reached 27 °C, at which point the deposit structure flowed and deformed seriously. On the other hand, evaporation losses had a strong effect on Low-viscosity ink. To reach the full potential of this promising technology, it is necessary to determine the effect of nozzle temperature on printing performance. This article provides a method for developing and applying Low-viscosity, Low-temperature food printing.

## 1. Introduction

Three-dimensional food printing is a popular production technology for creating customized and personalized food, and research on food printing is expanding in both academic and commercial fields [1,2,3]. There are three main types of food materials currently used for printing: (1) soft materials that can maintain their support after deposition; (2) Low-melting materials that can crystallize to form layered structures; and (3) polymeric materials with shear-thinning behaviors [4,5,6]. The viscoelastic and mechanical properties of gel materials are determined by changes in the rheological properties of the ink [7]. In addition, the temperature range from the barrel to the nozzle outlet is closely related to the rheological behavior of the ink. A bad temperature gradient will lead to poor deposition structure and support capacity [8,9,10]. Therefore, based on temperature regulation, it is the focus of further research on food printing to explore the influence of the temperature distribution of gel ink at the print head on printing performance. Gel food inks usually exhibit strong colloidal characteristics at low viscosities, and are generally less affected by food materials. Taking a gelatin solution as example, this paper explores the performance characteristics of Low-viscosity food inks during printing. The gelatin solution has the typical characteristics of a pseudoplastic fluid, with high-temperature melting and Low-temperature curing points which can be used for printing. It is often used as an additive to improve ink performance, edible film preparation and bioprinting [11,12,13]. Compared with other food gels, it shows greater fluidity at a low viscosity. Among materials used to study the influence of nozzle domain temperature on Low-viscosity food ink, it is easier to observe the flow behavior and more representative.

Low-temperature deposition manufacturing (LDM) is a cold-cure additive manufacturing technique (AM) suitable for Low-viscosity food inks [14]. In a typical printing process, the ink needs to be melted at the right temperature to be extruded through the nozzle. The temperature is raised by a heating device on the barrel wall, and the forming environment temperature is reduced via air cooling, water cooling or a cooling environment covering the entire printer [15,16,17]. The cylinder has a certain diameter, and the heat transfer is carried out through the cylinder and the nozzle in contact with the cooling environment. As a result, the temperature distribution is not uniform across the center, the surface of the cylinder and the nozzle. Due to the large temperature difference between heating and cooling, the lower temperature will be concentrated in the nozzle and its upper area. This is not conducive to the extrusion of ink and changes in temperature in the nozzle. Therefore, an innovative nozzle structure is needed to regulate the temperature gradient.

Temperature changes will affect the rheology and printing properties of temperature-dependent materials [18,19]. Gelatin is a highly temperature-sensitive material that exhibits more pronounced rheological behaviors at different temperatures [20,21,22]. It gels at low temperatures and melts at high temperatures, with a small range of variation that is easy to observe in experiments. A large number of studies have shown that the temperature changes in the barrel and the forming environment will affect the printing performance of the ink and can regulate the state of the deposition structure [15,16,23,24,25,26,27]. Proper structural design can improve the temperature gradient, which further affects the mechanical properties of printing [28,29,30,31,32]. The ink exhibits different forming capabilities and deformation effects under different nozzle outlet temperatures and temperature gradients. Therefore, it is important to understand the underlying mechanisms of different nozzle temperatures on Low-viscosity food inks. This contributes to the development and application of 3D printing food inks.

At present, the printing process of Low-viscosity food ink, especially the influence of nozzle temperature on printing performance, has not been fully understood. We adopted an integrated research approach, combining a numerical simulation and experimental validation to obtain more comprehensive and accurate research results. The effects of nozzle temperature on the printing properties, rheological properties and macroscopic and microscopic structures of Low-viscosity food inks were systematically studied. Through the results of this study, we aim to provide a scientific basis and optimization strategy for the printing process of Low-viscosity food inks. At the same time, our research also has important engineering value for the further development and application of food printing technology. In the following chapters, we will introduce our research methods, experimental design and an analysis of our results in detail. This will further verify and explain our novel discovery and reveal the key mechanism of the effect of nozzle temperature change on printing performance.

## 2. Materials and Methods

### 2.1. Material

To determine a suitable gelatin concentration, we printed ink with gelatin contents of 4, 6, 8, 10 and 12 wt%. According to Zhao’s master’s thesis (2019) and our actual printing conditions, 10 wt% was used in this study [33]. The gelatin (purchased from Henan Sugar Cabinet Co., Ltd., Henan, China, using 160 bloom) and water were prepared in a mixed solution at a ratio of 1:9, placed in a constant-temperature water circulation heating pot (purchased from 600 W Qun’an Experimental Instrument Co., Ltd., Zhejiang, China) for constant-temperature melting. It was defoamed (−2.5 MPa) using an air compressor (AP-01P, Puruiqi, Beijing, China) and stored at 40 °C.

### 2.2. Rheological Analysis of Gelatin

The rheological properties of the gelatin were characterized using a hybrid rheometer (Discovery HR-2, TA Company, Boston, MA, USA). The liquid gelatin was placed between a 25 mm parallel plate and the platform, with a gap of 1 mm. The setup was equilibrated for 1 min to reach the desired temperature (25, 30 and 35 °C), and excess meat was scraped off of the platform. The variance in the apparent viscosity with the shear rate was recorded as the shear rate was increased from 0.1 s^−1^ to 100 s^−1^. The elastic modulus G′ and the loss modulus G″ were calculated via a temperature sweep at a frequency of 1.6 Hz, and the samples were heated from 4 °C to 40 °C within 20 min. The variance in the apparent viscosity with temperature was recorded as the temperature was increased from 4 °C to 40 °C at a shear rate of 3.12 s^−1^. All measurements were conducted at an amplitude strain of 0.5% within the linear viscoelastic region (LVR) [9].

### 2.3. Configuration and Optimization of the Design of 3D Food Printing Equipment and Nozzle Structure

#### 2.3.1. Configuration of Food 3D Printing Equipment

A 3D printer for Low-temperature food deposition was designed, as shown in Figure 1. It added ambient cooling and nozzle domain temperature control to a traditional direct writing (DW) food printer. The ambient cooling consisted of a PID controller and a cooling box that surrounded the printer as a whole, as shown in Figure 1a. Nozzle domain temperature control was achieved with a customized nozzle structure via water circulation, as shown in Figure 1b.

A temperature sensor was installed in the nozzle to accurately control its temperature. However, nozzles used for food printing generally have diameters of less than 2 mm. Considering the structural size of the temperature sensor, it was impractical to install the temperature sensor inside the nozzle. Therefore, it was installed in the gap between the self-made print head structure and the nozzle, and the temperature of the water circulation recorded by the sensor replaced the temperature at the nozzle.

#### 2.3.2. Nozzle Structure Design

Convective heat transfer occurred between the cold air in the cooling environment and the nozzle. And the heat lost through the outer surface of the nozzle generated a thermal gradient. This phenomenon is particularly evident in Low-viscosity food printing because Low-viscosity foods are more sensitive to temperature changes. This will lead to a large deviation between the nozzle temperature and the barrel heating temperature, thus affecting the temperature control accuracy of the deposition process. Here, we designed an improved structure, as shown in Figure 1b. It was designed based on reducing heat loss and controlling temperature. The water flow temperature was heated via water circulation, and it was used to replace the thermal convection between the cold air and the nozzle. The simulation results in Section 4.1.1 show that the temperature distribution and control accuracy were improved.

The size of the optimized nozzle structure is shown in Figure 2. The inlet was above, and the outlet was below. In order to ensure a spiral flow of water around the nozzle, they were eccentrically designed. The water flowed from the inlet at a flow rate of 10 mm/s, ensuring the nozzle temperature during continuous rotation.

### 2.4. Finite Element Numerical Analysis

#### 2.4.1. Finite Element Simulation of Gelatin Fluid

Using the Heat Transfer and CFD module of COMSOL6.0 software (Comsol, Inc., Burling, MA, USA), the temperature field of the gelatin in the barrel and nozzle was analyzed via the finite element method.

#### 2.4.2. FEM Modeling of the Print Head

The better printing conditions obtained from the early orthogonal experiments were used for analysis. In order to accurately estimate the temperature distribution of the nozzle in the printing state, a steady-state thermal simulation model of the printing state was established based on the finite element method. Furthermore, geometric features such as chamfers and threads were ignored, and resistance wire heating was regarded as an isothermal shell heat source with a temperature of 30 °C. These decisions were made to reduce the difficulty of modeling during the analysis process and improve computational efficiency. The calculation grid and thermal boundary conditions of the nozzle domain are shown in Figure 3. The outer surface of the nozzle structure was simplified as a convective heat transfer boundary.

#### 2.4.3. Geometry and Boundary Conditions

To reduce the calculation time, the calculation range was limited to the nozzle and the barrel filled with gelatin (the part above the piston was ignored). The geometry simulation was constructed using Solidworks2018 (Solidworks, Inc., Dassault Systemes, Waltham, MA, USA). The mesh division of the simulated geometry is shown in Figure 3. Triangular mesh was used to discretize the fluid domain in the geometry. In order to ensure the calculation of turbulence, a finer physical field control grid was used. The grids of the entire model contained 3,542,847 units. The related boundary conditions are shown in Figure 3b. Flow was applied at the gelatin and water circulation inlets. The boundary conditions are listed as follows:

(1)Since it was assumed that there was no wall slip, the printing speed and water speed at the wall were zero;(2)The pressure at the outlet of the nozzle was 101,325 Pa, which was 1 atm;(3)The piston was considered a rigid body and did not deform. Therefore, the piston exerted a squeeze on the reservoir area at the same speed as its movement.

#### 2.4.4. Models and Assumptions

(1)Ink Flow Model

To analyze the flow characteristics of ink flowing through the nozzle, we evaluated the Reynolds number (*Re*_1_) of the shear thinning flow [34]:(1)$$Re_{1}=\frac{\rho_{1}d_{1}{}^{n}{\bar{v}}_{1}{}^{2-n}}{K{\left((3n+1)/(4n)\right)}^{n}8^{n-1}}$$
where *ρ*_1_ is the density, *d*_1_ is the nozzle diameter, ${\bar{v}}_{1}$ is the average velocity of gel ink, *n* and *K* are the flow index and consistency index (Pa·s^n^), respectively, derived from the rheological properties of the ink defined by the power law model [35]:(2)$$\eta \left(\dot{\gamma }\right)=m{\dot{\gamma }}^{n-1}$$
where *η* is the apparent viscosity (Pa·s) and $\dot{\gamma }$ is the local shear rate (s^−1^). The *Re* can be used to identify a flow regime in a pipe, such as a laminar flow (*Re* ≤ 2300), critical flow (2000 < *Re* < 4000), or turbulent flow (*Re* > 4000) [36]. The physical properties of the gel ink (90% moisture content) and the power law model coefficients are listed in Table 1.

For a given shear rate, the viscosity of the fluid is only related to *n*, and *n* quantifies the response of the apparent viscosity to the shear rate. *n* > 1, *n* < 1 and *n* = 1 indicated shear thickening, shear thinning and Newtonian fluid characteristics, respectively. The printing ink studied in this paper was a non-Newtonian fluid with *n* < 1. Both exponential coefficients in the power law model were measured via the rheological tests described in Section 2.2.

Due to the low Reynolds number, the incompressible gelatin fluid was under stable, isothermal and laminar flow conditions. The mathematical model of ink flow applied in this study included the following continuity and momentum conservation equations in the vector symbol [36]:(3)$$\rho_{1}\nabla \cdot u_{1}=0$$
(4)$$\rho_{1}\frac{\partial u_{1}}{\partial t}+\rho_{1}(u_{1}\cdot \nabla )u_{1}=\nabla \cdot \left[-pI+\mu_{1}\left(\nabla u_{1}+{\left(\nabla u_{1}\right)}^{T}\right)\right]+F$$
where **u**_1_ is the velocity vector, *p* is the hydrostatic pressure, **I** is the unit tensor, *μ*_1_ is the dynamic viscosity of the gel ink and **F** represents other external forces. The effect of gravity was neglected.

(2)Water flow model

To analyze the flow characteristics of the water circulation part, we evaluated the Re_2_ of the Newtonian fluid [37]:(5)$$Re_{2}=\frac{\rho_{2}{\bar{v}}_{2}d_{2}}{\mu_{2}}$$
where *ρ*_2_ is the density of water, ${\bar{v}}_{2}$ is the average velocity of water, *d*_2_ is the inlet diameter and *μ*_2_ is the dynamic viscosity of water. Table 1 shows that the *Re*_2_ of the water cycle was small and should indicate a laminar flow. However, the water flow had an obvious vortex flow in the improved nozzle structure, which was beneficial to the heat transfer between the water body and the nozzle. The model (Low-Reynolds-number k−ε turbulence model) applied in this study of water circulation included the following continuity and momentum conservation equations in vector symbols [38]:(6)$$\rho_{2}\frac{\partial k}{\partial t}+\rho_{2}u_{2}\cdot \nabla k=\nabla \cdot \left(\left(\mu_{2}+\frac{\mu_{2T}}{\sigma_{k}}\right)\nabla k\right)+P_{k}-\rho_{2}\varepsilon$$
(7)$$\rho_{2}\frac{\partial \varepsilon }{\partial t}+\rho_{2}u_{2}\cdot \nabla \varepsilon =\nabla \cdot \left(\left(\mu_{2}+\frac{\mu_{2T}}{\sigma_{\varepsilon }}\right)\nabla \varepsilon \right)+C_{\varepsilon 1}\frac{\varepsilon }{k}P_{k}-f_{\varepsilon }C_{\varepsilon 2}\rho_{2}\frac{\varepsilon^{2}}{k}$$
(8)$$P_{k}=\mu_{2T}\left(\nabla u_{2}:\left(\nabla u_{2}+{\left(\nabla u_{2}\right)}^{T}\right)-\frac{2}{3}{\left(\nabla \cdot u_{2}\right)}^{2}\right)-\frac{2}{3}\rho_{2}k\nabla \cdot u_{2}$$
where *k* is the turbulent kinetic energy, **u**_2_ is the velocity vector, *μ*_2T_ is the turbulent viscosity, *σ_k_* is the turbulent Prandtl number for *k*, *P_k_* is the production of turbulent kinetic energy, *ε* is the dissipation rate of turbulent kinetic energy, *σ_ε_* is the turbulent Prandtl number for *ε*, *f_ε_*, and *C_ε_*_1_ and *C_ε_*_2_ are constants. The effect of gravity was neglected.

(3)Assumptions
(1)Due to the slow extrusion speed and the small Reynolds number for food 3D printing, the laminar flow of the ink was adopted in the barrel;(2)The flow in the barrel was considered to be fully developed;(3)The influence of a small amount of gas in the barrel was ignored;(4)The ink did not slip on the wall during printing;(5)The friction between the piston and the storage cylinder was neglected;(6)The ink was a homogeneous material;(7)The ink was considered an incompressible material.


#### 2.4.5. Heat Transfer Settings

(1)Heat Transfer Analysis

The process of extruding gelatin ink involved natural convection heat transfer and contact heat transfer. The natural convection heat transfer refers to the heat exchange between the nozzle outlet, the barrel, the optimized nozzle structure and the external air. The contact heat exchange refers to the heat exchange between the ink, barrel wall, water circulation, nozzle structure, improved structure, heating sleeve and water circulation. There was no clear thermal contact resistance between the faces of the contact part, which was considered full-contact. No thermal contact resistance was set in the model, as shown in Figure 3.

(2)Heat source for barrel heating

The heating element was installed in the heat insulation cotton as a heat source during the experiment. And the temperature of the heating element was controlled via a closed-loop temperature sensor. The temperature of the cylinder wall at a distance of 13 mm from the bottom of the cylinder remained relatively stable at different temperatures (see Section 4.1.2). Therefore, the heating element and insulation cotton were regarded as a whole as a constant temperature boundary. The temperature of the entire module was defined as 30 °C (the average value of the ink in the cartridge measured at different temperatures) during the simulation.

(3)Convective heat transfer coefficient

Using *R_e_* and the Grashof number (*G_rL_*) (two dimensionless numbers), the type of heat transfer was determined as [37]:(9)$$G_{rL}=\frac{g\beta \left(T_{s}-T_{\infty }\right)L^{3}}{\nu^{2}}$$
where *g* is the acceleration of gravity, *β* is the volumetric thermal expansion coefficient, *L* is the characteristic dimension and *ν* is the kinematic viscosity of the air. *T_s_* and *T_∞_* are the temperatures of the surface air and the surrounding air. As the *G_rL_/Re*^2^ between the barrel, nozzle structure, heat insulation cotton and the cooling ambient air was much greater than 1, the heat transfer state between all models and the air was regarded as natural convection. The convective heat transfer coefficient (*h*) thus can be expressed as [29]:(10)$$h=\frac{0.59k}{L}G_{r_{L}}P_{r}$$
(11)$$P_{r}=\frac{\mu_{air}c_{p}}{k}$$
where *P_r_* is a temperature-related physical property (the Prandtl number), *μ_air_* is the dynamic viscosity of air, *c_p_* is the specific heat capacity and *k* is the thermal conductivity of air. *P_r_* was defined as a 15 °C cooling environment.

## 3. Printing Experiment and Result Verification

A Low-temperature deposition manufacturing (LDM) food printer was used for the experiments. According to the preliminary experimental results, suitable printing conditions were selected. The diameter of the nozzle was 1.0 mm, the layer height was 0.5 mm, the ambient cooling temperature was 15 °C, the heating temperature of the barrel was 55 °C, the water circulation temperature was 19, 21, 23, 25 and 27 °C and the travel speed of the nozzle was 1.56 mm/s. The standard printing process was carried out at refrigerated temperatures to evaluate the printing performance. We experimented with the initial and optimized structures of gelatin in different nozzle temperature domains.

An electron microscope (TD-4KHT, Sanqiang Teda) was used to analyze the deposition structure changes in the gelatin photographed. We selected the same location of the deposited structure for analysis, as shown in Figure 4.

## 4. Results and Discussion

In 3D printing Low-viscosity food, the rheological properties of the printed materials are significantly affected by temperature changes. In this section, the relationship between material printing performance and temperature is investigated and discussed by controlling the temperature of nozzle domain.

### 4.1. Analysis and Experimental Verification of Simulation Results

#### 4.1.1. Analysis of Simulation Results

Figure 5 shows the steady-state distribution of printing temperature for the initial and optimized structures. It can be seen from the figure that under the initial conditions, the temperature peaked at 30 °C in the middle of the inner wall of the barrel and bottomed at 19 °C at the outlet of the nozzle. The temperature drop at the top of the barrel was caused by the piston being affected by the external temperature. This was the same case in the optimized structure, but it had little impact on the printing performance because the ink would be heated again during the extrusion process. Moreover, a significant temperature drop occurred at the nozzle outlet due to the convective heat transfer between the cold forming environment and the nozzle. It formed a lower and steeper temperature gradient distribution, which was not conducive to nozzle temperature control during printing.

The temperature distribution of the nozzle under the optimized structure was more uniform. It had a gentler temperature drop than under the initial structure and a significant impact on the temperature of the nozzle. It can be seen that the temperature dropped only to a certain extent at the nozzle outlet. And there is a positive correlation with the water circulation temperature. This helps us indirectly characterize the nozzle outlet temperature using the water circulation temperature.

To compare the numerical difference between the simulated and actual printing processes, temperature measurements were carried out and are described in Section 4.1.2. Figure 6 shows the measured and simulated temperature curves at the nozzle outlet (Point 1) at three points under different conditions. The simulated and experimental temperatures at the nozzle outlet are also shown in the figure. And the deviations at different temperatures are 2.16, 1.66, 0.56, 0.02 and −0.26 °C, respectively. The average temperature deviation is about 0.83 °C. The mean difference from the actual value is 4.9%. Due to the simplified treatment of the model, the simulation results fit poorly at lower temperatures. Therefore, the simulation results are only used for further temperature distribution and control analysis.

#### 4.1.2. Experimental Verification

To verify the calculation accuracy of the simulation results and the temperature improvement ability of the optimized structure, several sets of experiments were carried out to measure the actual nozzle temperature at different water circulation temperatures. As shown in Figure 7, three-point temperatures of the initial and optimized structures were measured using temperature sensors (PT100 platinum resistance temperature sensor, ±(0.15 + 0.002 × |t|), where t is the temperature value to be measured).

Figure 6 shows the temperature data measured by the sensor and simulated by Comsol software. It can be seen from the figure that with the increase in the water circulation temperature, the growth rate at Point 1 was higher than at Points 2 and 3. The temperature of Point 1 even exceeded that of Point 2 at 23 °C. This was because the influence of the temperature of the water cycle weakens with increasing height and had a weaker effect at Point 2. In addition, there was little difference between Points 1 and 3 at 19 °C whether water circulation was present or not, but there was a gap of 8 °C at Point 2. The reason for this was the difference in the dominant factors under such a condition. In the absence of water circulation, the system was dominated by the heating device, and in the presence of water circulation, it was dominated by the water circulation temperature. Figure 6 also shows the deviation range between the nozzle outlet of the optimized structure and the water circulation temperature. It was 1.6, 1.8, 1.2, 1.5 and 1.2 °C, respectively. The average deviation was 1.46 °C, indicating that the optimized nozzle structure was more effective at temperature control, and there was a strong correlation between the water body and the nozzle outlet temperature, which is consistent with the simulation results. In this paper, the temperature of the water body, which can be easily controlled, replaces the temperature of the nozzle outlet for analysis.

### 4.2. Rheological Properties of Gelatin

Food materials used for extrusion molding must have an appropriate viscosity, ease of extrusion and a certain degree of adhesion to avoid deformation during the deposition of structures [39,40]. The apparent viscosity of the gelatin ink with a water content of 10 wt% is shown in Figure 8a. It decreased with the increase in the shear rate, so it was a typical pseudoplastic food printing material. In addition, the viscosity also inversely proportional to the temperature and decreased sharply in the range of 25–33 °C. This meant that gelatin ink had low degree of printability at a low temperature and a high degree of fluidity at a high temperature, which was not conducive to the structure-forming process of 3D printing, as shown in Figure 8b.

The storage modulus G′ and loss modulus G″ of the gelatin ink are shown in Figure 8c, which shows that they were temperature-dependent. G′ was significantly higher than G″ at low temperatures, showing a gel structure dominated by an elastic ability that made it easy to maintain the shape of the sedimentary structure [41]. With the increase in temperature, G′ and G″ gradually decreased, and the decline rate of G″ was gradually less than G′. When the gel network is depolymerized to a certain extent, the decrease rate of G′ slows down until it stops, and the decrease rate of G″ gradually exceeds G′. The change in internal tension led to the transfer of intermolecular and molecular internal forces, gradually weakening the elastic part of the gel structure and enhancing its ability to flow. The loss tangent (Tanδ = G′/G″) is a characteristic parameter used to describe viscoelastic behavior. A Tanδ less than 1 shows elastic characteristics, and a Tanδ greater than 1 shows viscous characteristics [42]. At temperatures higher than 32.25 °C, G″ was greater than G′, and the gelatin ink demonstrated a flow ability with liquid properties. In order to achieve successful printing, a reasonable temperature drive and precise rheological characteristics are needed.

Low-viscosity gelatin ink has an obvious fast gelling stage and a slow gelling stage. Upon cooling, gelatin undergoes physical gelation via the formation of triple helices. In the rapid gelation stage, the gelatin changes over time to show an initial rapid growth region in which a new helix is formed. In the slow gel phase, a slower growth region is present, involving spiral elongation. For gelatin gels formed in the fast-gelling regime, the rheological exponents depend on concentration but not on temperature. In addition, helical elongation occurs for a long time at each temperature and concentration for gelatin [43]. The slow gelling stage of 10 wt. % gelatin ink is large, and its slow gel time will be significantly greater than the rapid gel time. Since food printing is a rapid prototyping process, the slow gelation temperature is not suitable for printing analysis. Therefore, this paper used the rapid gelation temperature as a printing reference. It is calculated as follows [43]:(12)$$T_{l}(K)=\left(35+0.18c_{0}+273\right)\left(1-\frac{0.28}{2\ln c_{0}+5.06}\right)$$
where *c*_0_ is the initial concentration of gelatin (g/dL). Since the density of the ink is close to water, the initial value is 10 g/dL, and the calculated rapid gel temperature is 27.67 °C. Figure 8c shows the measured slow gelation temperature, 32.25 °C. These results are consistent with the study of Guo et al.

### 4.3. Macroscopical Effect of Nozzle Temperature on Printing Performance

#### 4.3.1. Rectangular Frame Sedimentary Structure

The deposition results of a rectangular frame with line segments lapped in the initial and optimized structure are shown in Figure 9. Gaps and less granular deposition are present in the initial structure. The optimized structure showed a serious extrusion issue at the nozzle domain temperature of 19 °C, and the material deposited on the transparent plastic film in a granular form. This was because the material gelatinized and blocked the nozzle tip at an ambient temperature that was too low. In addition, a simple extrusion behavior was observed at the nozzle temperatures of 21, 23, 25 and 27 °C, which was caused by the strong temperature sensitivity of the Low-viscosity material. The results show that the ink underwent a strong change in its printing performance in the 2 °C range.

However, as transparent plastic film is easy to fold and bend and is not easily made completely flat, the deposited structure will show flow gaps of different sizes on the film. This is somewhat beneficial for us to judge the flow properties of materials. With the gradual increase in temperature, there were an increasing number of flow gaps in the deposition structure on the film. It was not conducive to the deposition of the base structure in the printing process.

The differences in material properties between the actual and ideal print were attributed to the shear-thinning property of the material. As a result, the material which should show elastic ability at 19–25 °C demonstrated extrusion performance. A polymer is easier to extrude than a Newtonian fluid [44]. When the ink was sprayed into the nozzle, the viscosity declined under the condition of the shear force, making it easier to extrude. As a consequence, an ink that should demonstrate good extrusion performance above 27.67 °C (a fast-gelling temperature) can achieve a good deposition structure at 21 °C.

The flow of food materials in the nozzle was regarded as a Poiseuille flow [45]. The increase in temperature reduced viscosity but increased velocity, thus increasing the width of the deposition structure. However, the width of the deposition structure did not change significantly with the change in the nozzle temperature, as displayed in Figure 9, which was more obvious in Figure 4. This was because the viscosity of the printing material was too low and varied little with temperature at the same shear rate. This phenomenon indicated that the ink had a sudden change in response to temperature but had little effect on the width of deposition structure.

#### 4.3.2. Cylindrical Sedimentary Structure

Figure 10 shows the cylindrical deposition structure of gelatin ink at different nozzle domain temperatures (19, 21, 23, 25 and 27 °C) and the initial conditions. Among them, the initial structure and nozzle temperature at 21 °C showed similar printing results. This was due to the fact that the nozzle temperatures in the two states ware similar (see Section 4.1), and the inks exhibited nearly identical rheological properties. The printing performance of gelatin ink varied with the change in the nozzle domain temperature. The 19 °C sedimentary structure was granular and formed a complete cylindrical sedimentary structure at the base. After printing to a certain height, there was a granular aggregation due to the uneven support layer, and the particles collapsed to the side of the cylinder after excessive aggregation, which was caused by the low fluidity of the ink at 19 °C. At 21 °C, the granular structure suddenly disappeared, which was consistent with the analysis results in the previous section. At 21–27 °C, the porous defects in the gelatin gradually disappeared until it exhibited the best printing effect at 25 °C, and the flow phenomenon occurred due to strong fluidity at 27 °C. This was different from what was described in the above section because although a better rectangular frame deposition was presented, the fluidity was poor at 21 and 23 °C. The body structure deposition was prone to influence from the external environment, leading to pore defects. The fluidity of the ink was better at 25 °C, which made up for pore defects in the printing process, and it was suitable for printing conditions that did not require precise external environment control. Therefore, it is reasonably inferred that a more accurate ink model could be obtained at 21 and 23 °C when the external environment is well controlled. If the above material conditions and printing specifications were maintained, printing performance at 21, 23 and 25 °C varied under different environmental capabilities.

### 4.4. Microscopic Effect of Nozzle Temperature on Deposition Structure

The micro-deposition structures of the rectangular frame under the same conditions and position for the initial and optimized structures are shown in Figure 4. The deposition width at the horizontal symmetry line of the picture was regarded as the average width of the ink at this temperature. Gelatin shows inconsistent deposition widths on different surfaces (the hydrophobic plastic material or the rough, corrugated, stainless steel surfaces, etc.). This is caused by the combined effects of uneven surface morphology, the permeability difference of gelatin, and the drying and shrinkage of gelatin. A transparent plastic film is a typical hydrophobic plastic material with good transparency and a smooth surface. Therefore, the observed width is generally more consistent than the width observed on a rough surface in the gelatin printing process. In this paper, the deposition width of the gelatin was studied using a transparent plastic film as the substrate. The deposition widths at 19, 21, 23, 25 and 27 °C were 3.15, 3.73, 4.07, 4.05 and 4.20 mm, respectively. As can be seen from the microscopic map, dark characteristics appeared on both sides and bright characteristics appeared in the middle. This is because the light source illuminated by the electron microscope deflected at the ends of the arc section, resulting in lower levels of brightness at both ends. It can be more intuitively seen from Figure 11. After evaporation, the ink arc was reduced, both ends became flat, and brightness was increased. Under the initial conditions, there were obvious shadow-like lap marks in the middle of the sedimentary structure, and it had irregular contour edges compared to the optimized structures at 21, 23, 25 and 27 °C. The optimized structure at 27 °C had a smoother surface and better shape. The reason for this situation was that the initial condition had no nozzle domain water circulation heating structure. The actual temperature at the nozzle was far lower than the charging barrel temperature but close to ambient cooling temperature. The viscosity of the gelatin material sharply increased with the decrease in temperature, which led to a decline in the passing ability of the ink at the nozzle and thus a poor deposition effect. This phenomenon was more significant at 19 °C after optimization and was accompanied by a serious granular deposition structure. In addition, lap marks gradually weakened, and edges became smoother with the increase in temperature in the range of 21–27 °C. The best effect emerged at 27 °C, but the viscosity was too low, and an obviously uneven flow appeared on the test plate, as shown in Figure 12.

The normal extrusion of gelatin ink was achieved under these conditions except that the deposition had obvious grainy quality at 19 °C. There was little difference in the deposition width, which was consistent with the conclusion in Section 4.3. The mutability of Low-viscosity gelatin to a temperature response was further confirmed. Considering the overall structure, lap marks and edge effects, the optimized 23 and 25 °C demonstrated better printing performance. This phenomenon illustrated the importance of nozzle domain temperature control for the overall and microscopic deposition performance in the 3D food printing process.

### 4.5. Effect of Evaporation on Deposition Structure of Low-Viscosity Ink

In this study, the deposition structure of Low-viscosity ink showed obvious evaporation loss and shrinkage behavior within 10 min (room temperature environment) after printing, as shown in Figure 11. This is relatively common in gelled 3D printing materials but usually occurs after a long time or in extreme environments [46]. With the increase in time, the structure width of the gelatin ink decreased after evaporation. The small-scale deformation due to the flow disappeared, showing a flatter frame structure. This phenomenon was caused by the low molecular content of the Low-viscosity ink. The reduced molecular content extended the gelation time of the gelatin ink and made it more prone to melting at room temperature. This was significantly different from conventional high-concentration food printing. Although small-scale deformation and the disappearance of lap marks were beneficial to the appearance and perception of the deposition structure, the deposition width shrank by about 55%, which was not conducive to the formation of the next layer of the structure or accurate control of the deposition structure. In the printing process, the temperature should be maintained as low as possible to inhibit the influence of evaporation loss on the printing structure. The volumetric evaporation loss of the structures that needed to be preserved at room temperature was considered in the construction of the 3D model.

## 5. Conclusions

The temperature in the cylinder has an important impact on the printing performance and deposition structure of extrusion 3D printing with Low-viscosity food. In order to better understand the variation mechanism and potential role of the temperature domain from the barrel to the nozzle, we developed a nozzle structure to quantify the temperature of the nozzle domain and carried out a thermal analysis simulation via the finite element method, as well as experimental verification. Through simulation and experimental comparison, a 10 wt. % gelatin ink demonstrated an ideal printing performance when exposed to varying environmental control capabilities at nozzle domain temperatures of 21 °C, 23 °C and 25 °C. In the presence of external interference, the nozzle domain temperature should be set as 25 °C to ensure excellent printing performance. The actual printable temperature is about 6 °C lower than the temperature guided by the rheological properties. The response of print performance to temperature changes was abrupt. It mutated at 19–21 °C, and the ink deposition was granular below 19 °C and normal above 21 °C. With the increase in temperature, the lap traces between the lines weakened, and the porous defects in the body structure decreased. However, it showed a strong flow capacity at 27 °C, and the flow deformation of the structure was serious. In addition, evaporation loss has a great effect on the deposition structure of the Low-viscosity ink. When stored at room temperature for 10 min, the evaporation loss occurred obviously, and the deposition width decreased by about 55%. This paper provides a certain reference for forming or printing other Low-viscosity food at low temperatures. Further research can be carried out to test the printing performances of different inks at low temperatures or on different surfaces. Future research should explore the forming performances and rheological properties of more Low-viscosity inks at low temperatures and expand the application range of Low-viscosity ink in food printing.

## Figures and Tables

**Figure 1 foods-12-02666-f001:**
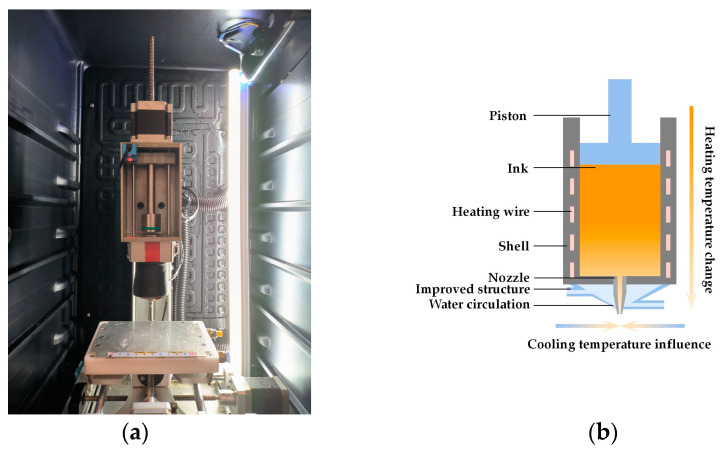
Configuration of 3D food printing equipment: (**a**) custom-designed Low-temperature food deposition 3D printer; (**b**) custom-made print head structure.

**Figure 2 foods-12-02666-f002:**
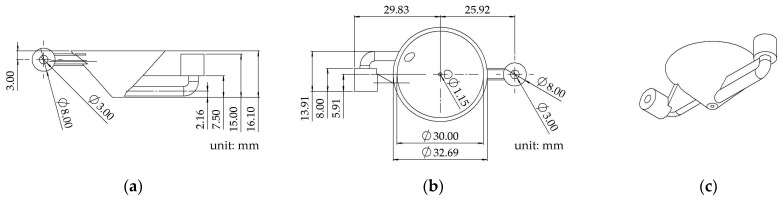
The size diagram of the optimized nozzle structure: (**a**) front view; (**b**) vertical view; (**c**) equiaxial side view.

**Figure 3 foods-12-02666-f003:**
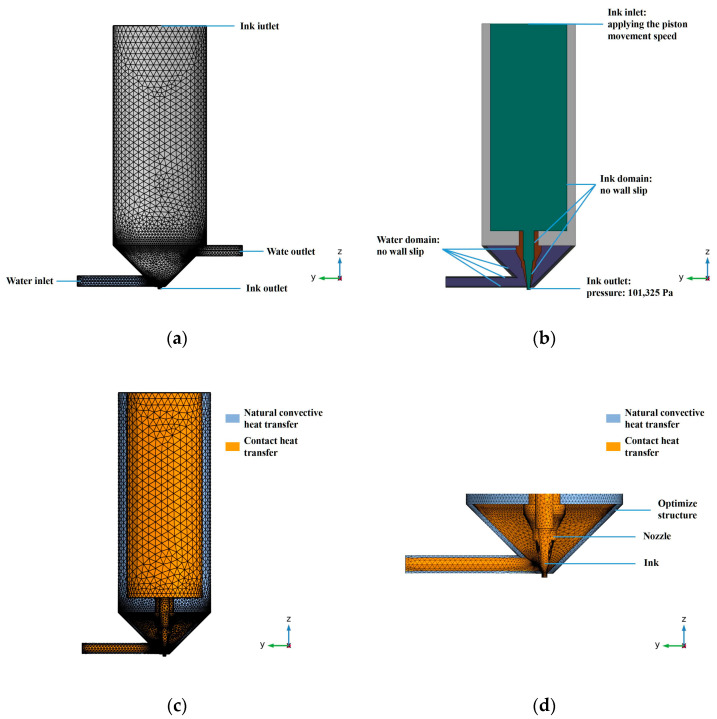
Schematic diagram of 3D printer nozzle and print head: (**a**) meshing of the overall model; (**b**) the essential boundary conditions; (**c**) thermal boundaries of the overall model; (**d**) meshing and heat transfer boundaries of the nozzle structure region.

**Figure 4 foods-12-02666-f004:**
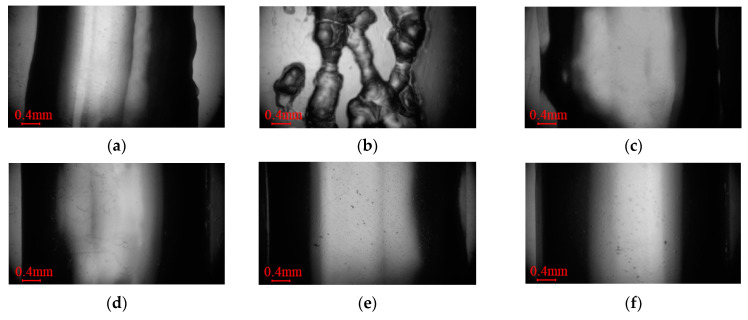
Micrograph of sedimentary points of the unified model: (**a**) initial structure; (**b**) optimized structure at 19 °C; (**c**) 21 °C; (**d**) 23 °C; (**e**) 25 °C; (**f**) 27 °C.

**Figure 5 foods-12-02666-f005:**
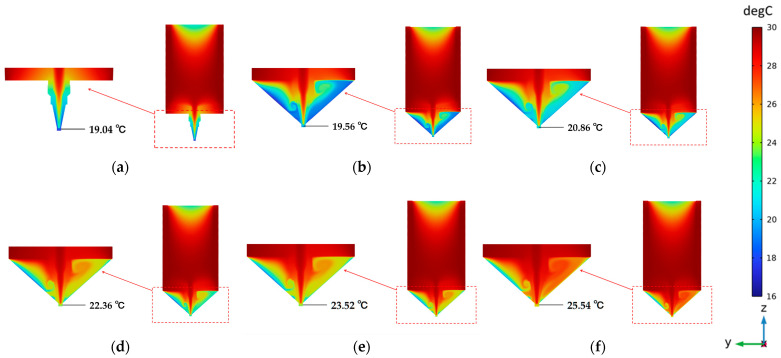
Steady-state temperature distribution under the initial and optimized structures: (**a**) initial structure; (**b**) optimized structure at 19 °C; (**c**) 21 °C; (**d**) 23 °C; (**e**) 25 °C; (**f**) 27 °C.

**Figure 6 foods-12-02666-f006:**
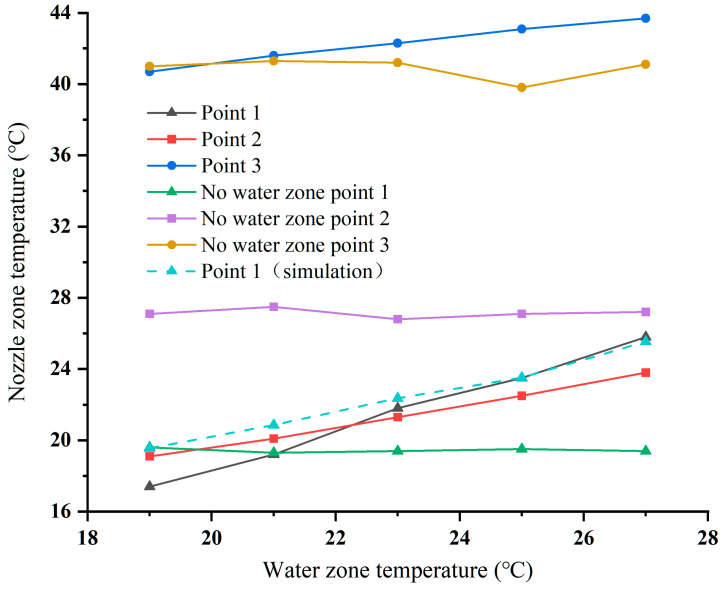
The temperature data measured by the sensor at three points and the simulated temperature at Point 1.

**Figure 7 foods-12-02666-f007:**
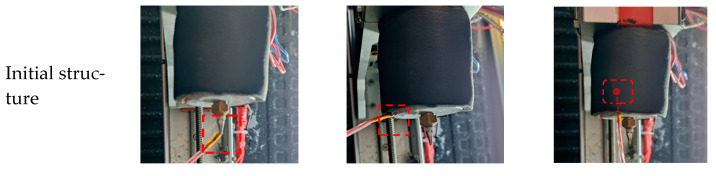
Temperature measurement at three points (the red box indicates the measurement location).

**Figure 8 foods-12-02666-f008:**
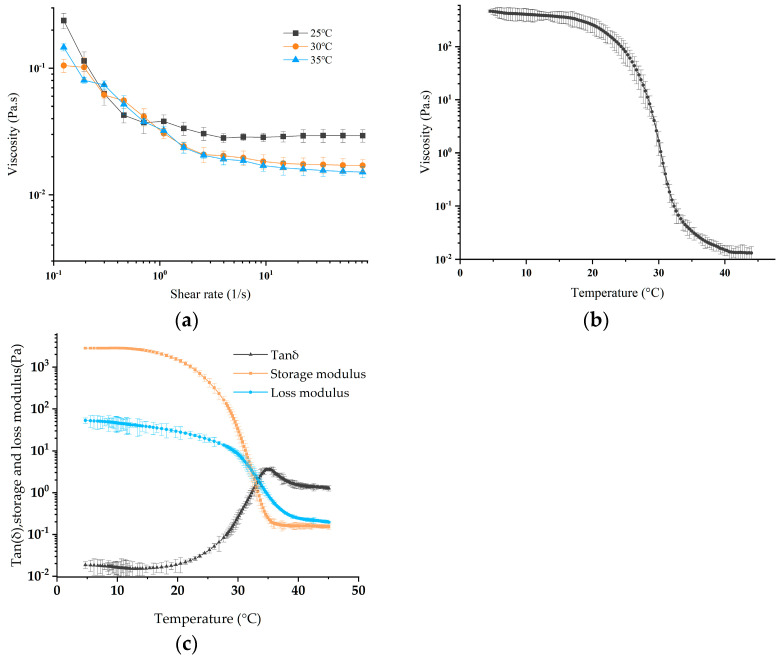
(**a**) The apparent viscosity of gelatin ink with a water content of 10 wt% at different temperatures as a function of shear rate; (**b**) the apparent viscosity of gelatin ink with a water content of 10 wt% as a function of temperatures; (**c**) storage modulus and loss modulus of gelatin ink with a water content of 10 wt% at different temperatures.

**Figure 9 foods-12-02666-f009:**
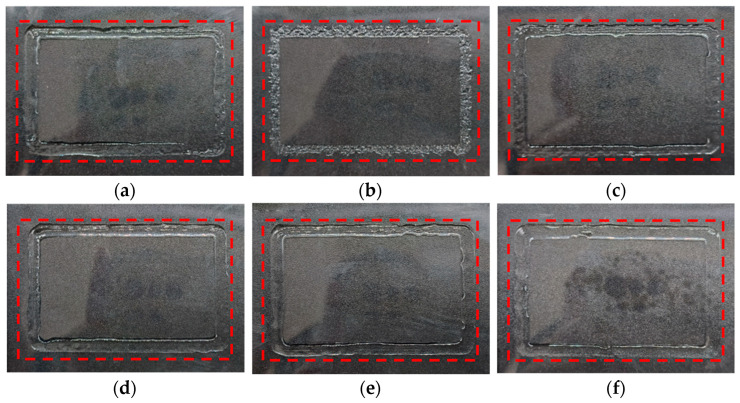
Initial and optimized deposition structure of rectangular frame at different nozzle domain temperatures: (**a**) initial structure; (**b**) optimized structures at 19 °C; (**c**) 21 °C; (**d**) 23 °C; (**e**) 25 °C; (**f**) 27 °C.

**Figure 10 foods-12-02666-f010:**
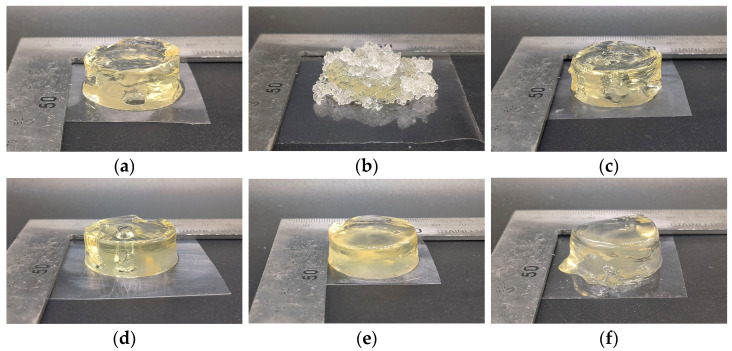
Cylinder deposition structure of initial structure and optimized structure at different nozzle domain temperatures: (**a**) initial structure; (**b**) optimized structure at 19 °C; (**c**) 21 °C; (**d**) 23 °C; (**e**) 25 °C; (**f**) 27 °C.

**Figure 11 foods-12-02666-f011:**
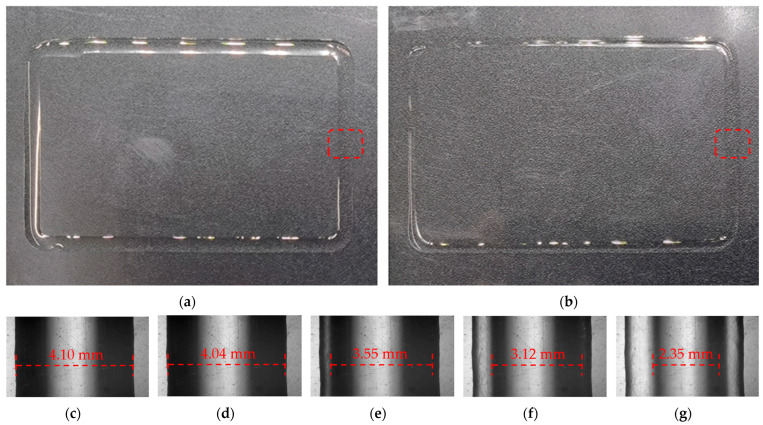
Deposition structures at different time under the optimal structure condition at 25 °C: (**a**) normal view at 2 min; (**b**) 10 min; (**c**) micro view at 2 min; (**d**) 4 min; (**e**) 6 min; (**f**) 8 min; (**g**) 10 min.

**Figure 12 foods-12-02666-f012:**
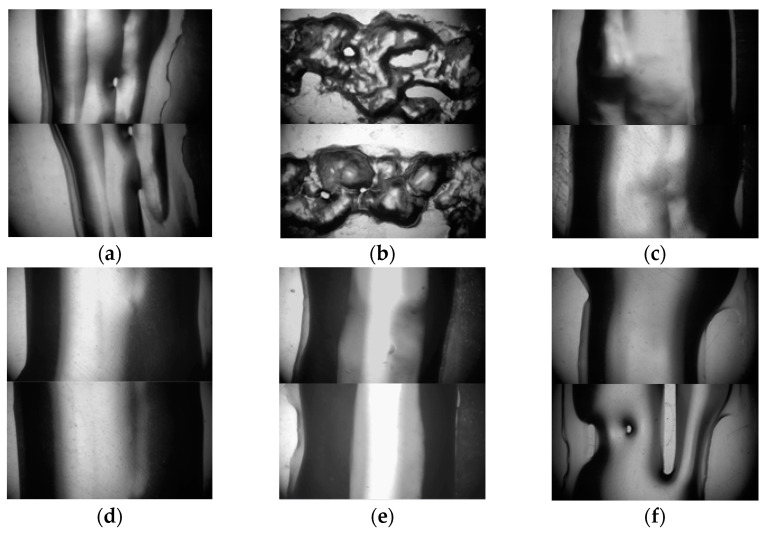
Typical bad spots and flows of deposition structures at different temperatures: (**a**) initial structure; (**b**) optimized structure at 19 °C; (**c**) 21 °C; (**d**) 23 °C; (**e**) 25 °C; (**f**) 27 °C.

**Table 1 foods-12-02666-t001:** The geometric parameters of FEM.

Parameter	Symbol	Value	Unit
The density of the gel ink	*ρ* _1_	1	g/cm^3^
The nozzle diameter	*d* _1_	1	mm
The average velocity of the gel ink in the nozzle	${\bar{v}}_{1}$	0.83	mm/s
The consistency index	*K*	0.04	Pa·s^n^
The flow index	*n*	0.58	
Reynolds number of the gel ink	*Re* _1_	0.03	
The density of water	*ρ* _2_	1	g/cm^3^
The inlet diameter	*d* _2_	3	mm
The average velocity of water	${\bar{v}}_{1}$	10	mm/s
The dynamic viscosity of water	*μ* _2_	10^−3^	Pa·s
Reynolds number of water	*Re* _2_	30	
The Prandtl number of air	*P_r_*	0.713	

## Data Availability

The data presented in this study are available upon request from the corresponding author.

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
