# Peer review of "The Effect of Nozzle Temperature on the Low-Temperature Printing Performance of Low-Viscosity Food Ink"

_foods, 2023, doi:10.3390/foods12142666_

Round 1
Reviewer 1 Report
Overall comments: The following comments were suggested to improve the manuscript.
Specific comments:
1. Line 103, What is ‘DW’ food printer?
2. Line 113-115, can this accurately measure nozzle temperature?
3. Line 131-132, more details about model development should be provided. Which module was used, initial and boundary conditions, mesh, governing equations….
4. Line 224, the model should be verified first, then the simulation result can be used for further analysis.
5. What type of temperature sensors, accuracy?
6. Line 263, all the results& discussion should be included in this section, right? A significant amount of results were presented at earlier sections.
7. In figure 7, are these values measured, if so, the standard deviation should be added.
Reviewer 2 Report
The manuscript “The effect of nozzle temperature on the low-temperature printing performance of low-viscosity food ink” has been written well. The comments to improve the manuscript are mentioned below:
1. Abstract: Need to be revised. The concluding statement should show the utilisation and application of low-temperature food printing.
2. Lines 11 to 13: Does not make any sense. Rewrite the sentence
3. Introduction: Line 40: What do you mean by influence trend???
4. Material: Don't write words like "we printed". A standard protocol with any citation or authenticated source needs to be given for the preparation of gelatin ink. Why gelatin and ink in the ratio 1:9 were used? On what basis this optimisation is done?
5. Line 93-98: Why citation for rheological analysis of gelatin is not given?
6. Results and discussion: Results and discussion are presented and explained well. However, some similar findings can be added to support the data.
7. Figures are presented well
8. Recheck the whole manuscript for typing and grammatical errors.
The English is satisfactory however, please check the entire manuscript for typing and grammatical errors.
Reviewer 3 Report
The present submission reports data from the finite element method, using COMSOL software, on the temperature distribution around a nozzle for gelatin deposition and compares the results with actual experimental data. Although an extensive and laborious work, the submission needs improvements.
A major issue is the language, which at specific sections is not good, while other text sections are quite understandable and easy to follow. Nevertheless, the manuscript should be edited and improved regarding the language.
In addition, a few faults are throughout the text, highlighted and commented on in the attached PDF. Below, I provide two examples. The rest can be found in the edited PDF.
-There are data in Figure 7 from phase transition measurements (oscillatory temp. sweeps) that are not reported here. The authors should include all the measurement procedures, like the continuous shear temp. ramp in Figure 7(b).
- Is this particular approach that the width of the gelatin deposit is only related to the nozzle temperature too simplistic? For example, if the substrate surface was a hydrophobic plastic material or a rough, corrugated, stainless steel surface, the observed width will remain the same?

The manuscript should be edited and improved regarding the language.
Reviewer 4 Report
In the paper, the influence of nozzle temperature on the printing performance, rheological properties and microstructure of low-viscosity food ink using numerical simulation and experimental research has been studied.
The paper is interesting, however, mostly of an engineering value. This is an engineering project. The authors should expand the scientific elements of their studies. The authors should indicate the novelty elements of their study and try to explain the obtained results. Only such an explanation determines the scientific nature of the research performed.
There is lack of short and clear statements on the novelty of this research. The authors should try to generalize the results of their research, not just report them.
The paper is not clearly written:
1. Figure 1b is not clear:
- Print head is not indicated,
- The nozzle is not indicated,
- Subscription is not clear: “Interaction between each temperature domain and self-made print head structure” – where are “each temperature domains” ?
2. What does it mean: “The optimal printing conditions obtained from the initial experiments were used for analysis”: what is the criterion of optimization ?
3. Figure 2 is not clear:
- Where are the thermal boundary conditions shown ?
4. “2.4.4. Geometry parameter settings”:
- Where are these parameters ?
5. Equations 2-4 are not clear:
- Eq. 2 is wrong,
- What is u ?
- What is grid number ?
- Prandtl number is not defined,
- What way Eq. 4 has been obtained ?
- L is characteristic dimension (not length),
- Dimensions of all parameters should be given.
Your paper is hard to read.
Round 2
Reviewer 1 Report
The authors have addressed all the comments in this revised version.
Author Response
Thank you to the reviewers for reviewing our papers and providing valuable advice.
Reviewer 3 Report
The authors have applied the suggested improvements and changes.
The revised manuscript, in it's current format, can be accepted for publication.
Wishes for wide acceptance by your readers.
Author Response

(The authors gave the same response as above.)

Reviewer 4 Report
- The boundary conditions are not shown in Figure 3
Do you understand the term boundary conditions ?
- Eq.2: tau not sigma, gamma dot not gamma,
- Prandtl number is not defined,
- What way Eq. 10 has been obtained?
Language has been partly improved.
